# Biobased chiral semi-crystalline or amorphous high-performance polyamides and their scalable stereoselective synthesis

Paul N. Stockmann[1], Daniel Van Opdenbosch[2], Alexander Poethig[3,4], Dominik L. Pastoetter [1], Moritz Hoehenberger[1], Sebastian Lessig[1], Johannes Raab[1], Marion Woelbing[1], Claudia Falcke[1], Malte Winnacker[3,4], Cordt Zollfrank[2], Harald Strittmatter[1] & Volker Sieber [1,2,4]*

The use of renewable feedstock is one of the twelve key principles of sustainable chemistry. Unfortunately, bio-based compounds often suffer from high production cost and low performance. To fully tap the potential of natural compounds it is important to utilize their functionalities that could make them superior compared to fossil-based resources. Here we show the conversion of (+)-3-carene, a by-product of the cellulose industry into ε-lactams from which polyamides. The lactams are selectively prepared in two diastereomeric configurations, leading to semi-crystalline or amorphous, transparent polymers that can compete with the thermal properties of commercial high-performance polyamides. Copolyamides with caprolactam and laurolactam exhibit an increased glass transition and amorphicity compared to the homopolyamides, potentially broadening the scope of standard polyamides. A four-step one-vessel monomer synthesis, applying chemo-enzymatic catalysis for the initial oxidation step, is established. The great potential of the polyamides is outlined.

[1] Fraunhofer IGB, Bio, Electro and Chemocatalysis BioCat, Straubing Branch, Schulgasse 11a, 94315 Straubing, Germany. [2] Technical University of Munich, Campus Straubing for Biotechnology and Sustainability, Schulgasse 16, 94315 Straubing, Germany. [3] Department of Chemistry, Technical University of Munich, Lichtenbergstr. 4, 85748 Garching, Germany. [4] Catalysis Research Center, Technical University of Munich, Ernst-Otto-Fischer-Straße 1, 85748 Garching, Germany. *email: sieber@tum.de

Polyamides (PA) are an important class of high-performance polymers and have been used in various industries (automotive, textile, medical, etc.) since the first industrial polyamide PA-6.6 was developed by Carothers in 1936 (refs. [1,2]). They are either produced by condensation of diacids with diamines, leading to AABB-type polymers, or amino acids for the AB-type, respectively. This type can alternatively be produced by ring-opening polymerization (ROP) of lactams. The most famous example for a lactam used as a monomer is ε-caprolactam, a seven-membered cyclic amide which was first polymerized to PA-6 by Schlack in 1938 (ref. [3]). It is usually produced from fossil oil-based cyclohexane after the oxidation to cyclohexanone, oxime formation, and subsequent Beckmann rearrangement. Recently, bio-based ways to synthesize caprolactam starting from glucose and fructose via hydroxymethylfurfural (HMF) have been described[4,5]. Due to the increasing awareness of dwindling fossil resources and environmental problems that arise using fossil oil as basis for plastics, alternative monomer resources for polymers in general and for polyamides specifically have become a major focus of research[6–19]. Sustainable monomer sources can be serious alternatives to fossil oil, as demonstrated by the commercially available bio-based polyamides PA1010, PA11, or PA410. These examples are linear, non-chiral condensation polyamides composed of diacids or amino acids derived from castor oil. In this context, we consider especially monoterpenes from renewable feedstocks to be a promising source for biogenic polymers. Monoterpenes such as limonene, camphor, menthone, α- and β-pinene, or 3-carene provide valuable carbon structures such as aliphatic rings and can be isolated from waste streams of biomass-utilizing processes in high volumes[20]. The latter three are the main components of turpentine oil with a combined annual production volume of about 350 kt per year, primarily isolated from the kraft pulping process (sulfate turpentine, 200 kt) or by distillation of resins extracted from conifers (gum turpentine, 100 kt). The composition highly depends on the species and origin of the utilized conifers. In Southeast USA, α-pinene (60–75%) and β-pinene (20–25%) are more common, whereas turpentine from Scandinavia and Russia contains considerable amounts of (+)-3-carene (40%)[21]. Chemically functionalized terpenes used for polyolefins, polyesters, polycarbonates, polyacrylates, and others with promising properties have been reported, underlining their enormous potential for bio-polymers[22–37]. However, terpene-based polyamides are still rare. The first terpene-based bio-polyamide was synthesized by Hall[38] in 1963 via the cationic ROP of a β-pinene based lactam. Recently, Winnacker et al.[39,40] reproduced, investigated, and optimized the synthesis and polymerization and tested the application of the new polyamide in cell growth control. Another approach by the same group was the oligomerization of a menthone-derived lactam[41,42]. For both the commercial polyamides, such as PA6, PA11, PA12, and the lactam-based bio-polyamides, factors limiting broader fields of application remain: relatively low glass transition temperatures ($T_g$) and melting temperatures ($T_m$) or low molecular weights ($M_n$ and $M_w$) in addition to a costly or unscalable synthesis. However, the terpene-based polyamides possess thermal properties in the range of high-performance polymers. We recently described the synthesis of 3R-caranlactam from (+)-3-carene and its polymerization to poly-3R-caranamide[43]. This new polyamide has interesting thermal properties and a molecular weight in the range of commercial PA6 and PA12, but the monomer synthesis was challenging and involved toxic and expensive chemicals.

In this work, we report the synthesis of the new diastereomeric 3S-caranlactam and an optimized synthesis for 3R-caranlactam. 3S-caranlactam is a methyl group diastereoisomer of 3R-caranlactam with considerable different properties and polymerizability. The isomers are selectively prepared by an epoxide-ketone rearrangement and a suitable kinetic or thermodynamic control of the intermediates. A facile and straightforward one-vessel synthesis of 3S-caranlactam in a 4.0 L reactor is presented. Over four steps, an overall yield of 25% is reached. The monomers are polymerized to poly-3R-caranamide and poly-3S-caranamide, and the co-polymerization of 3S-caranlactam with caprolactam (CL) and laurolactam (LL) is also performed. The polymerizations are investigated regarding reaction time, temperature, and other factors. A crucial effect of the amount of applied activator is detected, which is in accordance with polymerization theory. Thermal properties of the homo- and co-polyamides are characterized by differential scanning calorimetry (DSC). Bio-based semi-crystalline or amorphous polyamides and co-polyamides with unique high-performance thermal properties are obtained. The amorphicity of poly-3R-caranmide and several co-polyamides is underlined by the preparation of transparent, sonlvent-cast films. A crystal structure of the semi-crystalline poly-3S-caranamide is also presented. As a conclusion, the great potential of these polyamides is shortly outlined.

## Results

**Monomer synthesis.** Compared to reported literature, the initial oxidation of the double bond of (+)-3-carene (**1**) is achieved by an epoxidation instead of the challenging alcohol synthesis by hydroboration[45,46]. In pathway A (Fig. 1), application of immobilized Cal-B lipase, an industrial enzyme from *Candida antarctica*, or buffered diluted peracetic acid exclusively leads to epoxide **2-3S** (where "S" is the configuration of the stereo-centre of C3)[47–49]. The metal free enzymatic method generates epoxides in high yields under mild conditions, can be conducted in green solvents such as ethyl acetate, and prevents the potentially dangerous aggregation of peracetic acid. In both cases, the epoxidation—which is rather uncommon—was only little exothermic and therefore easily controlled. The yield after distillation was over 80%. As a next step, the rearrangement of the epoxide to a ketone was performed. A literature protocol using high amounts of $ZnBr_2$ in EtOAc resulted in 59% yield, consisting of a mixture of the diastereoisomeric ketones **3-3S** and **3-3R**, which can be identified by Gas chromatography mass spectrometry (GCMS), (Supplementary Methods, Supplementary Fig. 1, Supplementary Table 1)[50]. A diastereoselective rearrangement has not been reported to date. As a mixture of stereoisomers would eventually result in an atactic polyamide, we were interested in a stereoselective catalysis which would enable the synthesis of the pure isomers—but complex[51,52] and costly[53–56] catalysts should be avoided. We assumed that a concerted mechanism leads to inversion of the methyl group, whereas an ionic two-step mechanism results in a mixture[53,57–59]. After screening for optimized reaction conditions (Supplementary Tables 2–8) with respect to solvent polarity, concentrations of reactants, and temperatures, we identified several trends: (I) with decreasing polarity of the solvent, inversion of the methyl group to isomer **3-3S** is preferred; (II) increasing substrate concentration leads to formation of high-boiling molecules, presumably oligomers; (III) oligomer formation decreases at higher temperatures; (IV) various side products are formed under aqueous acidic conditions; and (V) very acidic conditions under exclusion of water are most promising.

The solvent polarity is crucial for the regioselectivity and the stereoselectivity. Hydrocarbons are most suited, whereas application of polar ethers, such as THF, and nucleophilic alcohols drastically reduce the yield (Supplementary Table 2). The role of the anion is also important, as the reaction was comparably unsuccessful with other iron salts ($FeCl_3 \cdot 6H_2O$: 22% ketone selectivity, Supplementary Table 3, entry 9; $Fe(OAc)_2$: no

**Fig. 1 Monomer synthesis overview.** Synthesis pathway with yields and diastereomeric excess (de) of intermediates for the production of the lactam isomers **5-3S** and **5-3R** (yields refer to the small-scale experiments from purified starting material). The labelling of the stereo-centre C3 at all intermediates follows the recommendation for terpene carbon skeleton numbering of M. W. Grafflin, which suggest that the initial carbon labels of (+)-3-carene are fixed also in case of functionalization[44].

conversion, Supplementary Table 4, entry 4). $Zn(OTf)_2$ also gave satisfying results (Supplementary Table 4, entry 2); however, the reaction time was long even at comparably high catalyst concentration. A control reaction with $NH_4(ClO_4)$ showed no conversion. Alternatively, sulfonic acids in cyclohexane or toluene are suitable (Supplementary Table 5). The combination of non-polar solvents and very strong acids can also be applied for the stereoselective rearrangement of other epoxides and will be investigated in more detail in the future. For the synthesis of **3-3R**, **2-3R** was synthesized via the bromohydrine and base-induced epoxide formation, following an adjusted protocol of limonene epoxide production[11]. Application of the $Fe(ClO_4)_2 \cdot H_2O$/cyclohexane system gave isomer **3-3R** in excellent selectivity (de 95%). However, the work-up by distillation led to isomerization and 18% of **3-3S** were formed. As **3-3S** is the kinetic product, whereas isomer **3-3R** is thermodynamically favoured, an isomerization under protic acidic conditions (Supplementary Table 9) is possible and the equilibrium is 80:20 in favour of **3-3R**[45,60]. An allyl alcohol was identified as intermediate by GCMS (Supplementary Fig. 1). The synthesis of the oximes **4-3S** and **4-3R** was realized by conversion with hydroxylamine hydrochloride in over 80% yield. In both cases, the *trans*-oxime was the major isomer. Surprisingly, the reaction time was considerably longer for **3-3R** than for the S-isomer. We used that observation to produce **3-3R** from the 3R-enriched equilibrium mixture—thereby making the use of NBS obsolete—with an isomeric purity of over 97%: Addition of small amounts of hydroxylamine hydrochloride led to selective conversion of **3-3S** to **4-3S** in the presence of **3-3R**, which was then separated by distillation (Supplementary Fig. 2). The overall yield starting from (+)-3-carene (**1**) was 47%. Initial experiments indicated that the formation of acetals by the reaction of **3-3S** with alcohols such as glycerol could be used accordingly. The last step of the synthesis was the Beckmann reaction of **4-3S** and **4-3R** to the corresponding lactams 3S-caranlactam (**5-3S**) and 3R-caranlactam (**3-3R**). The yield was over 70% in both cases. As reported before for **5-3R**[43], the selectivity with respect to the nitrogen insertion was over 90%, and crystallization from EtOAc afforded pure **5-3S** or **3-3R**, respectively. Surprisingly—as other Lewis acids were proven unsuitable[43]—some of the perchlorate Lewis acids showed potential for a catalytic Beckmann rearrangement of **4-3S**, with the optimum reaction conditions being investigated at the moment.

We then scaled the reaction to 2.50 mol—or 1.25 mol for the enzymatic epoxidation—of **1** in a 4.0 L reactor and conducted the synthesis as a one-vessel process (Fig. 2, Supplementary Methods, Supplementary Figs. 3 and 4). As only washing steps and solvent changes were required, all intermediates remained in the reactor throughout the whole process. The crystallization was partly achieved in the reactor at 15 °C (approximately 50%), and the formed crystals could be filtered effortlessly. The remaining product was then isolated by crystallization at −20 °C separately. The crystallization protocol was not optimized so far, and a complete crystallization within the reactor might well be possible. In general, the crystallization of **5-3S** was not affected by the accumulated side products (e.g. cymene and unidentified high-boiling aliphatic compounds). The yield after recrystallization was 24% over four reaction steps. **5-3S** was synthesized accordingly with 20% overall yield; the enriched **3-3R** had to be purified separately by vacuum distillation and was then re-transferred to the reactor for the oximation. The process will be upscaled to the 100 L reactor shortly.

Although the synthesis is not fully optimized, several sustainable aspects are worth mentioning. The presented process requires only moderate reaction conditions and no elaborate reaction equipment. Only little amounts of metal are used during the process. In addition—as no low-boiling, interfering side products are formed—cyclohexane is retrieved during the process and can be reused; EtOAc and MeCN can also be recovered from the mother liquor by distillation after product crystallization. Finally, the product purity that is required for polymerization is reached by crystallization, avoiding material- or energy-consuming methods. However, for a fully sustainable synthesis, the amount of washing solutions—which were used in great excess so far—must be reduced, a catalytic method for the Beckmann rearrangement needs to be implemented, and an increase of the overall yield is required.

**Preparation and investigation of the homopolyamides.** As mentioned before, **5-3R** polymerizes by anionic ROP under application of NaH as initiator and in situ generated N-acetylated **5-3R** to poly-3R-caranamide (**poly5-3R**), reaching an average molecular weight number ($M_n$) over 30 kDa[43]. The addition of $Ac_2O$ during this process is challenging to control and to reproduce due to reaction temperatures reaching values above the boiling point of $Ac_2O$ and monomer sublimation. Therefore, we used N-benzoyl-3R-caranlactam (**Bz5-3R**) and N-benzoyl-3S-caranlactam (**Bz5-3S**,) as solid, high-boiling activators (Fig. 3). Gel permeation chromatography (GPC, Supplementary Methods,

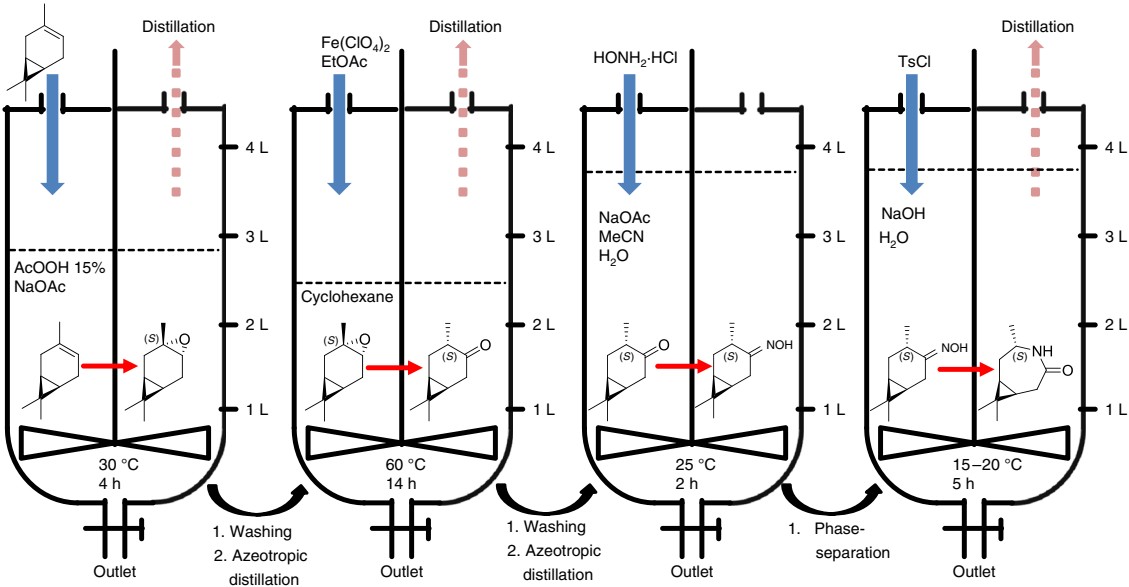

**Fig. 2 Up-scale of 5-3S.** Schematic one-vessel reaction cascade in a 4.0 L scale for the synthesis of lactam **5-3S**. (1) Addition of **1** to the epoxidation reagent AcOOH, washing, addition of cyclohexane and subsequent azeotropic distillation; (2) Meinwald rearrangement of **2-3S** to **3-3S** with Fe(ClO₄)₂, washing, and solvent exchange to MeCN; (3) oximation to **4-3S** using a buffered solution of HONH₂·HCl; and (4) Beckmann rearrangement to **5-3S** in basic media with tosyl chloride (TsCl).

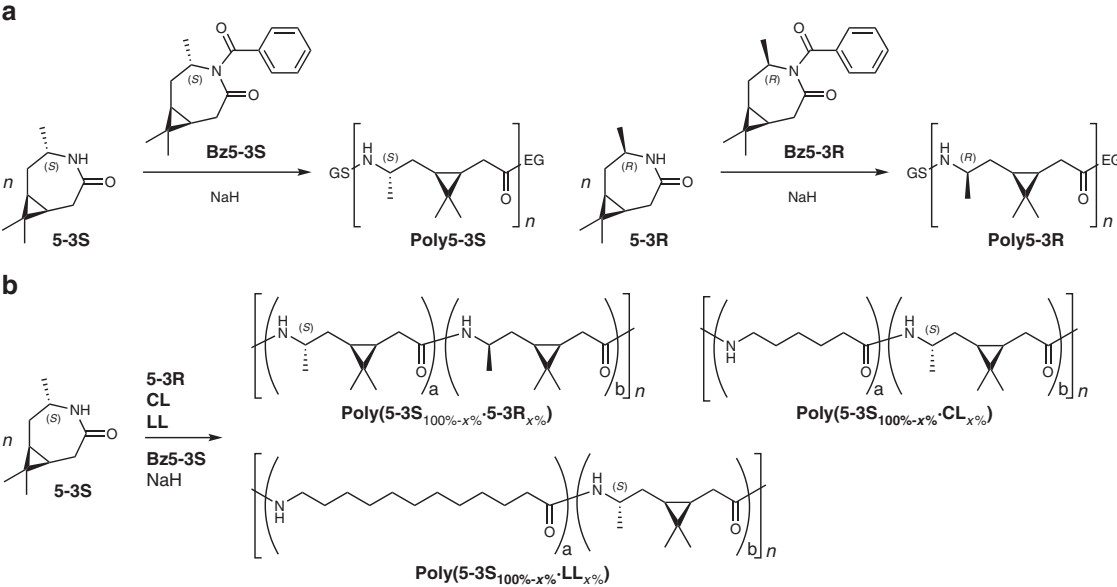

**Fig. 3 Polymerization overview.** Anionic ring-opening polymerization of the (+)-3-carene based lactams **5-3S** and **5-3R** (**a**) and co-polymers of the general structures poly(5-3S$_{100\%-x\%}$·5-3R$_{x\%}$) from **5-3S** and **5-3R**, poly(5-3S$_{100\%-x\%}$·CL$_{x\%}$) from **5-3S** and CL, and poly(5-3S$_{100\%-x\%}$·LL$_{x\%}$) from **5-3S** and LL (**b**).

Supplementary Fig. 5, Supplementary Table 10) was chosen for determination of the molecular weight as measurements of **poly5-3R** and **poly5-3S** with MALDI-TOF did show the characteristic peak distance of 167 $m/z$, but only oligomers up to 6.5 kDa could be detected (Supplementary Fig. 6). This is a known phenomenon for polyamides[39,61]. The initial polymerization reactions of lactam **5-3S** were carried out in an evacuated glass vial equipped with a metal screw cap with a rubber septum, a magnetic stir bar, NaH on paraffin wax as an initiator, and **Bz5-3S** as an activator (Polymerization method A, Fig. 4, Supplementary Fig. 7). The

activator concentration was varied, and the effect of the reaction temperature was evaluated at 180 and 220 °C.

At 180 °C, an $M_n$ of over 10 kDa and an $M_w$ over 16 kDa are observed, whereas at 220 °C the values do not exceed 7.5 kDa. For both temperatures, the decreasing amount of **Bz5-3S** results in an increasing molecular weight, as expected. However, this effect is stronger at 180 °C and $M_n$ increases over 70%; at 220 °C, only 35% average chain growth is observed. At a starting monomer/activator ratio (ratio$_{A/M\ start}$) of approximately 50, the rise of the molecular weights attenuates considerably in both cases. The polydispersity

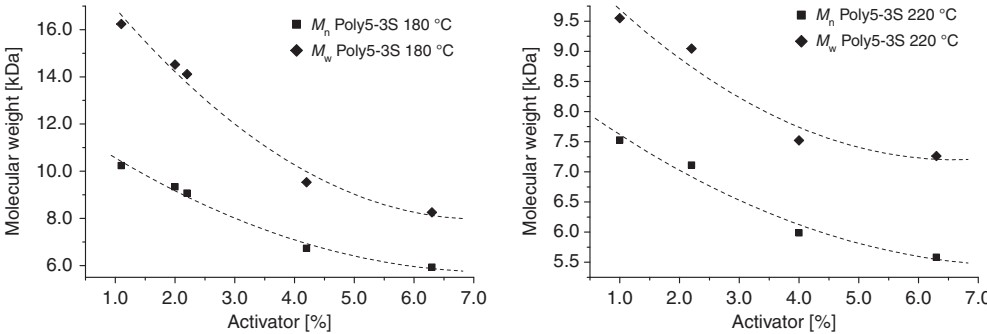

**Fig. 4 Polymerization of 5-3S.** Effect of the reaction temperature and the activator concentration on $M_n$ and $M_w$ (Supplementary Figs. 8 and 9, Supplementary Table 11). Conditions: 1.8 mmol **5-3S**, 2.0–5.5 mol% NaH on paraffin, 180 or 220 °C, 1 h. Molecular weights refer to masses over 1.0 kDa (GPC).

index (PDI) is between 1.2 and 1.6 for all ratio$_{A/M\ start}$, being a typical value for ROP which usually yields polymers with a narrow mass distribution[62]. DSC revealed $T_g$s of 111–115 °C and $T_m$s of 245–285 °C (Supplementary Fig. 10, Supplementary Tables 11 and 12) for a reaction temperature of 180 °C, whereas the melting onset was as low as 230 °C for a reaction temperature of 220 °C. To remove the residual monomers and oligomers, the polyamides were grinded to powders and stirred in a water/ethanol solution for several hours (Polymer work-up method A). Though this process was successful, it became clear that the conversion of **5-3S** could not be calculated from the isolated polymer yield, as uncontrolled polymer losses during the process (filtering, grinding, etc.) are unavoidable. The yield of isolated poly-3S-caranamide (**poly5-3S**) was 60–80%. In addition, sublimation of the monomer during the polymerization was detected particularly at long reaction times, which distorts the monomer/activator ratio. Similar challenges were observed during the polymerization of lactam **5-3R**.

Therefore, we changed the polymerization set up and used a heating block covered with an aluminium foil to guarantee homogeneous temperature inside the polymerization vial and used nitrogen instead of vacuum for inert reaction conditions—this prevented sublimation almost completely (polymerization method B, Supplementary Fig. 17). To determine the conversion without the described drawbacks, **poly5-3S** and **poly5-3R** were dissolved in hexafluoro-2-isopropanol (HFIP) in the polymerization vial directly after the reaction. After complete dissolution of the polymer, samples of the homogeneous solution were analysed by GPC and NMR to measure the ratio of unreacted and incorporated monomer (Fig. 5). In both cases, the NMR and GPC analysis were in good agreement. NMR revealed that no isomerisation of the methyl group or side reactions of the three-membered ring occurred (Supplementary Figs. 18 and 19). This is worth mentioning, as the defined relative (and absolute) configuration gives rise—as indicated by chemical logic and demonstrated for other chiral lactams[39]—to chiral polyamides. **Poly5-3R** was formed more readily and at lower activator concentrations than **poly5-3S**; 0.3 mol% activator was enough to surpass 80% conversion of **5-3R**, whereas 2.5 mol% were required for **5-3S**. The lactams reached a conversion of almost 90%, which is surprisingly high for substituted lactams, especially as aliphatic side-chains are attached at the β- and γ-position[63]. Substituents in these positions usually set the Gibbs energy to less negative values and lead to ring closure, therefore lowering the reactivity. Consequently, aliphatic substituents such as methyl- or propyl groups in the γ-position lead to an unfavourable polymer–monomer equilibrium and decreased conversion[10,63,64]. However, although several examples have been published, detailed predictions about the polymerizability and polymer–monomer equilibrium of bicyclic lactams are challenging[65–67]. For lactam **5-3S**, no significant

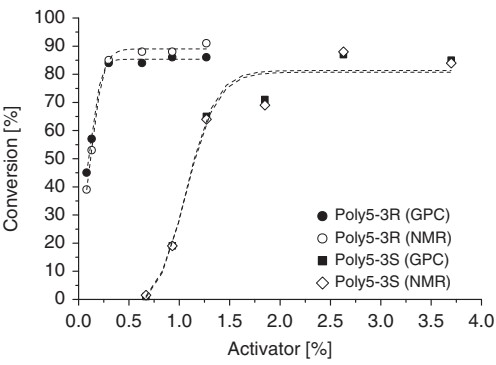

**Fig. 5 NMR and GPC investigations of the monomer conversion.** Conversion of the lactams and illustrative sigmoidal Boltzmann Fit depending on an increasing amount of activator **Bz5-3S** determined by NMR and GPC (Supplementary Figs. 11–16, Supplementary Table 13). Conditions: 3.0 mmol **5-3S** or **5-3R**, 3.0 mol% NaH on paraffin, 190 °C, 1 h.

conversion was observed beneath a ratio$_{A/M\ start}$ of approximately 1:100, whereas even a ratio$_{A/M\ start}$ as small as 1:1200 (<0.1%) lead to a conversion of over 40% within 1 h for the stereoisomeric lactam **5-3R**. If 1% activator was applied, the polymerization was completed in seconds. The $M_n$ of **poly5-3R** and, at high conversion levels, the $M_n$ of **poly5-3S** increased at decreasing amounts of the activator. However, below a certain activator concentration and conversion, the observed molecular weights are low for **poly5-3S** (Fig. 6).

As the formation of **poly-5S** was considerably slower than of **poly-5R**, we investigated the conversion at a given activator concentration (3.3%) at different reaction times at 180 °C.

The conversion reached a maximum of 80% after 2 h; 94% of that was completed after about 50 min (Supplementary Fig. 20, Supplementary Table 14). As reported by us previously, **poly5-3R** did not possess a melting point in the DSC analysis (Supplementary Methods, Supplementary Table 15, Supplementary Fig. 21)[43]. We presume that the high degree of aliphatic substitution prevents the polymer chains to crystallize, even at long tempering times. This has not been reported for terpene-based polyamides before, but has been reported for other alkyl-chain-substituted short-chain bio-polyamides[10]. DSC of **poly5-3S** revealed that the polyamide is semi-crystalline with a comparably weak $T_g$ at around 105 °C and a $T_m$ of up to 280 °C, surpassing the $T_g$ and $T_m$ of unsubstituted PA6 by 50 and 60 °C, respectively. The decomposition temperature under nitrogen atmosphere was around 360 °C. The high $T_g$s are caused by the three-membered ring in the polymer backbone and the resulting reduction in one degree of rotational freedom in the

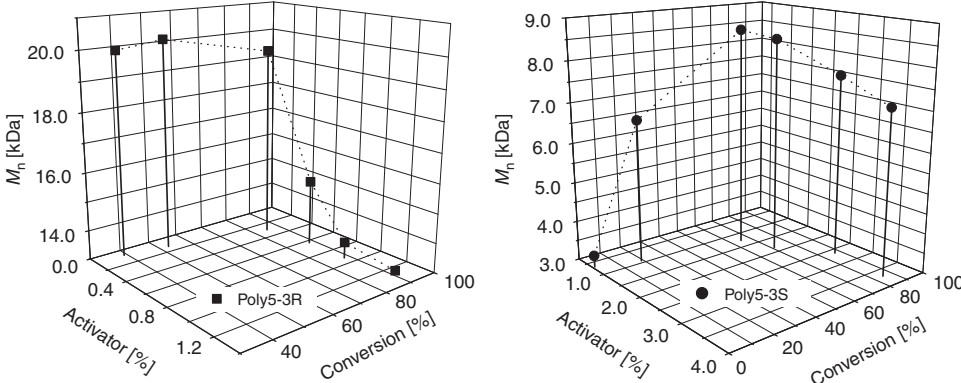

**Fig. 6 Effects of the amount of activator.** Influence of the activator concentration on the conversion and $M_n$ of **poly5-3R** and **poly5-3S** (Supplementary Figs. 11–14, Supplementary Table 13). Conditions: 3.0 mmol **5-3S** or **5-3R**, 3.0 mol% NaH on paraffin, 190 °C, 1 h.

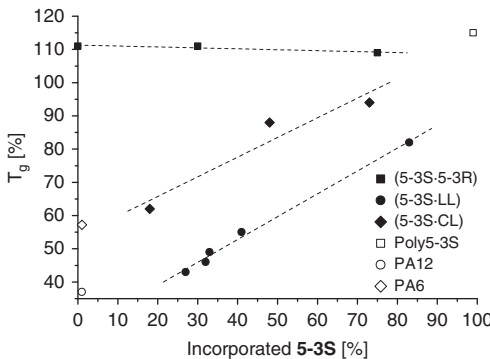

**Fig. 7 Co-polymerization effects.** Influence of the inclusion of **5-3S** on the $T_g$ of co-polyamides with CL and LL (Supplementary Figs. 23–27, conditions: Supplementary Table 16).

polymer backbone. To rule out the possibility that the lower molecular weight enables the, therefore, shorter polymer chains to form crystals, a short-chain **poly5-R** ($M_n = 10$ kDa; Supplementary Fig. 22) was synthesized and analysed by DSC—still, no melting was observed. As no other differences in the monomer or the polymer chain are present, this finding can directly be attributed to the diastereomeric methyl group that, consequently, causes not only a different ring structure in the lactam but also strong structural changes in the polymer chain. To support these findings, we investigated the monomers and both polyamides with XRD (Supplementary Methods, Supplementary Notes 1 and 2).

**Preparation and investigation of the co-polyamides**. As the three-membered ring of **5-3S** might decrease the segmental chain motion[68], we hypothesized that the random inclusion of this motif in the more regular chains of PA6 and PA12 could not only result in an increased $T_g$ but also in a decrease or even complete loss of long-range order. Apart from that, the copolymerizability of **5-3R** and **5-3S** was tested (Fig. 3b).

The co-polymerization was successful by applying similar reaction conditions as for the homopolyamides and could be verified by NMR. In all cases a complete consumption of **5-3S** was not achieved. If equimolar amounts of **5-3S**, CL or LL were used, the integration in the backbone was 96% (CL) and 89% (LL). The $T_g$ of the co-polyamides with CL and LL was shifted to higher temperatures with increasing amount of **5-3S** (Fig. 7). For example, a built-in of 48% in the regular PA6 chain in **copoly(5-3S$_{48\%}$·CL$_{52\%}$)** resulted in a $T_g$ of 88 °C, whereas the $T_g$ of **copoly**

**(5-3S$_{41\%}$·LL$_{59\%}$)** is shifted to 55 °C (Supplementary Table 16, entries E10 and E7). In both cases, no melting point was observed in DSC and the polymers were clear, yellow blocks. If only 18% of **5-3S** were integrated in the PA6-backbone, a broad melting area between 160 and 190 °C caused by cold crystallization (110–150 °C, Supplementary Fig. 26 i) was observed, indicating that this amount is not enough to completely suppress the establishment of a long-range order. **Copoly(5-3S$_{83\%}$·LL$_{17\%}$)** has a $T_g$ of 82 °C and a melting range of 220–250 °C. The broad melting ranges suggest that **5-3S** is not regularly distributed in the polymer chain and that the different reaction kinetics of the monomers lead to a gradual increase of the slow-reacting lactam in the growing chain throughout the polymerization reaction. As only one melting range is observed, the parallel formation of the homopolyamides PA6/12 and **poly5-3S** can be excluded[69]. It is worth mentioning that the $T_g$ seems to increase in a linear manner in a certain incorporation-range of **5-3S**, approximately between 25% and 80%. We hypothesize that outside of this range, effects arising from crystallinity affect the $T_g$. As the crystallinity is drastically reduced within the range, the increasing $T_g$ can be directly attributed to the amount of **5-3S** in the polymer backbone.

Investigation of the co-polymerization of 1:1 mixtures of **5-3S** and CL/LL revealed that the monomers were consumed at different rates (Supplementary Figs. 28 and 29, Supplementary Table 17). The co-polymerization of **3-5S** and CL almost exclusively starts with the conversion of CL before **3-5S** is also incorporated. Some examples for intermediate **poly(5-3S·CL)** were: **poly(5-3S$_{18\%}$·CL$_{72\%}$)** at 60 s (6.7% total conversion), **poly(5-3S$_{37\%}$·CL$_{63\%}$)** at 240 s (41% total conversion) and **poly(5-3S$_{49\%}$·CL$_{51\%}$)** at 60 min (82% total conversion). The co-polymerization with LL proceeded vice versa, and the co-polymer composition was **poly(5-3S$_{66\%}$·LL$_{34\%}$)** at 60 s (7.9% total conversion), **poly(5-3S$_{58\%}$·LL$_{42\%}$)** at 240 s (34.5% total conversion) and **poly(5-3S$_{52\%}$·LL$_{48\%}$)** at 60 min (64.4% total conversion). From this, it can be concluded that the relative rates of consumption are CL > **5-3S** > LL.

To further evaluate the optical properties, we fabricated films of the co-polyamides and **poly5-3R** by dissolving them in HFIP and slow evaporation of the solvent. Commercial and self-made PA6 and PA12 resulted in colourless non-transparent films. From the amorphous co-polymers, however, relatively transparent foil-like films could be produced (Fig. 8). Although some of the films show little amounts of residual monomers and—due to the preparation method—inclusions of solvent and other irregularities, the increasing transparency is obvious.

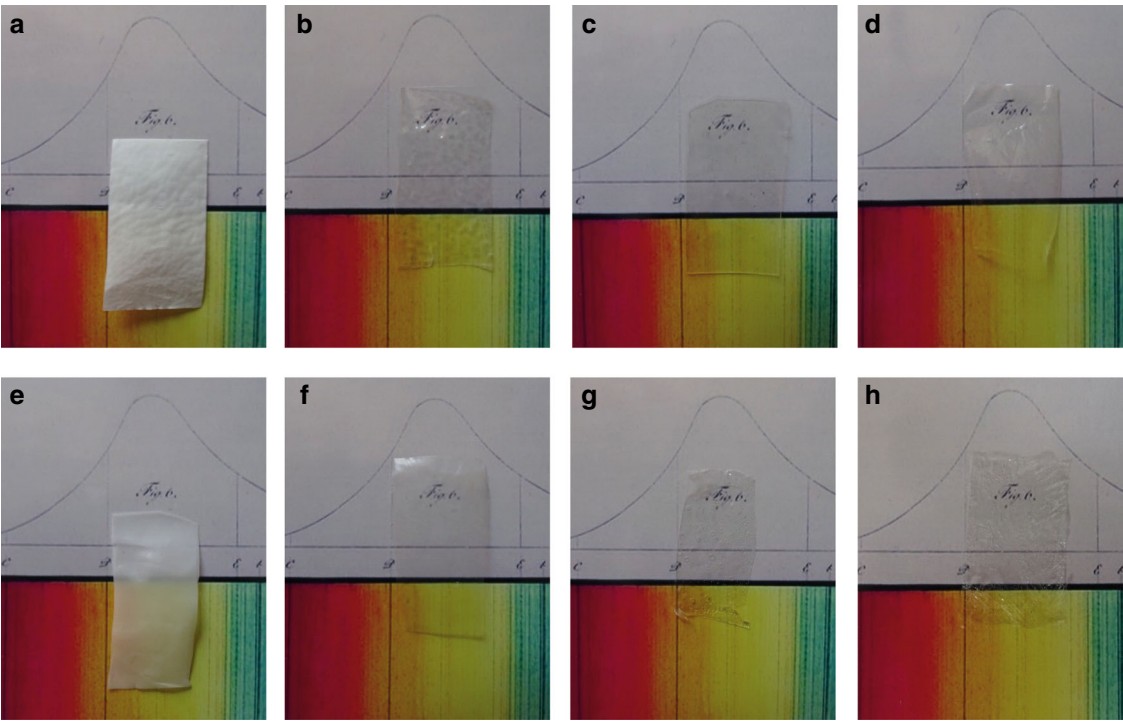

**Fig. 8 Transparent polyamide films.** Photographs taken under identical conditions of pure polymers and co-polymers in front of a sector of the Fraunhofer lines. **a** PA12; **b** copoly(5-3S$_{32\%}$·LL$_{68\%}$); **c** copoly(5-3S$_{33\%}$·LL$_{67\%}$); **d** copoly(5-3S$_{41\%}$·LL$_{59\%}$); **e** PA6; **f** copoly(5-3S$_{18\%}$·CL$_{82\%}$); **g** copoly(5-3S$_{48\%}$·CL$_{52\%}$); **h** poly5-3R.

**Crystal structures of the monomers.** Comparison of the structures of the monomers, determined by single-crystal X-ray diffractometry (details see Supplementary Note 1), shows that the methyl-group inversion at C3 leads to a changed conformation of the seven-membered ring. This might explain the different properties of the monomers: **5-3S** has a higher melting point (about 170 °C instead of 140 °C as observed for **5-3R**, Supplementary Fig. 30); increased monomer sublimation was detected during the heating process, and **5-3S** crystallizes more rapidly from EtOAc.

The higher quality low-temperature (100 K) data obtained for **5-3R** (CCDC 1938733) agree with published values for a room temperature measurement (CCDC 145220)[46]. The crystal structures derived from the diffraction patterns (Fig. 9, Supplementary Fig. 31a, b, Supplementary Table 18) illustrate that hydrogen bonds are formed between symmetry-identical molecules in **5-3S** (CCDC 1938732), as well as between the two independent molecules in the asymmetric unit of **5-3R**. In case of **5-3R** a dimer formation via classic hydrogen bonding through antiparallel arrangement of the amide groups is observed and non-classic hydrogen bonding to the methyl groups at C9 and C10 leads to an extended hydrogen bonding through the crystal. The independent molecules of **5-3R** have identical absolute configurations (Supplementary Figs. 32–34). In case of **5-3s** the bonding network is different, since each amide oxygen atom forms hydrogen bonds to an amide hydrogen atom of one symmetry generated molecule, as well as a non-classic hydrogen bond to one of the C5-bound hydrogen atoms of another symmetry generated molecule in the crystal structure. Comparison of the lactam structures with their respective energy-minimized shapes (MOPAC, PM7, singlet) yields the root mean square displacements 0.032 Å (S) and 0.042 Å (R). This confirms the expected low-energy conformation of the unconstrained monomeric species. The amide motif (C3–N1–C4–C5) is almost planar for both isomers, reflecting

the partial double bond character. In contrast, the absolute configuration of the stereogenic centre C3 significantly influences the different conformations of the seven-membered ring. In case of **5-3R**, derived from the thermodynamically favoured ketone **3-3R**, the seven-membered ring can be divided in two groups of coplanar arranged atoms: one plane spanned by C1–C2 and C5–C6 and the other by C2–C5 including N1. In case of **5-3S** the situation is different since one plane is spanned by C3–C5 including N1 and the other plane by C5–C6 and C1–C3.

**Crystal structure of poly5-3S.** The diffraction patterns recorded from **poly5-3S** were characteristic of a semi-crystalline polymer (Supplementary Data 1, Supplementary Fig. 31c), allowing structural determination by the direct space method simulated annealing (SA, Supplementary Methods, Supplementary Note 2, Fig. 10, Supplementary Figs. 35 and 36). By scaling the amorphous reference patterns to match them at 2Θ values outside the range of Bragg peaks, we obtained a fraction of crystalline phase ($f_c$) of 0.42 from the ratios of the integrated intensities. Patterns from **poly5-3R** showed no distinct reflexes, except those from residual monomers (Supplementary Fig. 31d).

In **poly5-3S**, the arrangement of N1 and O1 within the polymer chain is similar to that found in β-peptides[70]. Within crystallites of **5-3S**, hydrogen bonds are formed between pairs of antiparallel chains (Fig. 10, Supplementary Fig. 35). Consequently, the obtained crystal structure for **poly5-3S** is one of antiparallel two-strand β-sheets. The inter-sheet distances are $a/2 = 4.90(3)$ Å, and the repeat distances along the chain $c = 6.44$ (6) Å (Fig. 10, Supplementary Table 18). The distance from N1 to O1 of 2.79 Å corresponds well to the distances typically found in polyamides[71]. Further, the density of the crystalline phase of 1.167 g cm$^{-3}$ is a good match for the densities of the **5-3S** monomer crystalline phase of 1.147 g cm$^{-3}$ and well in the range of standard polyamides[2]. Finally, the bond angle C5–C6–C1 of

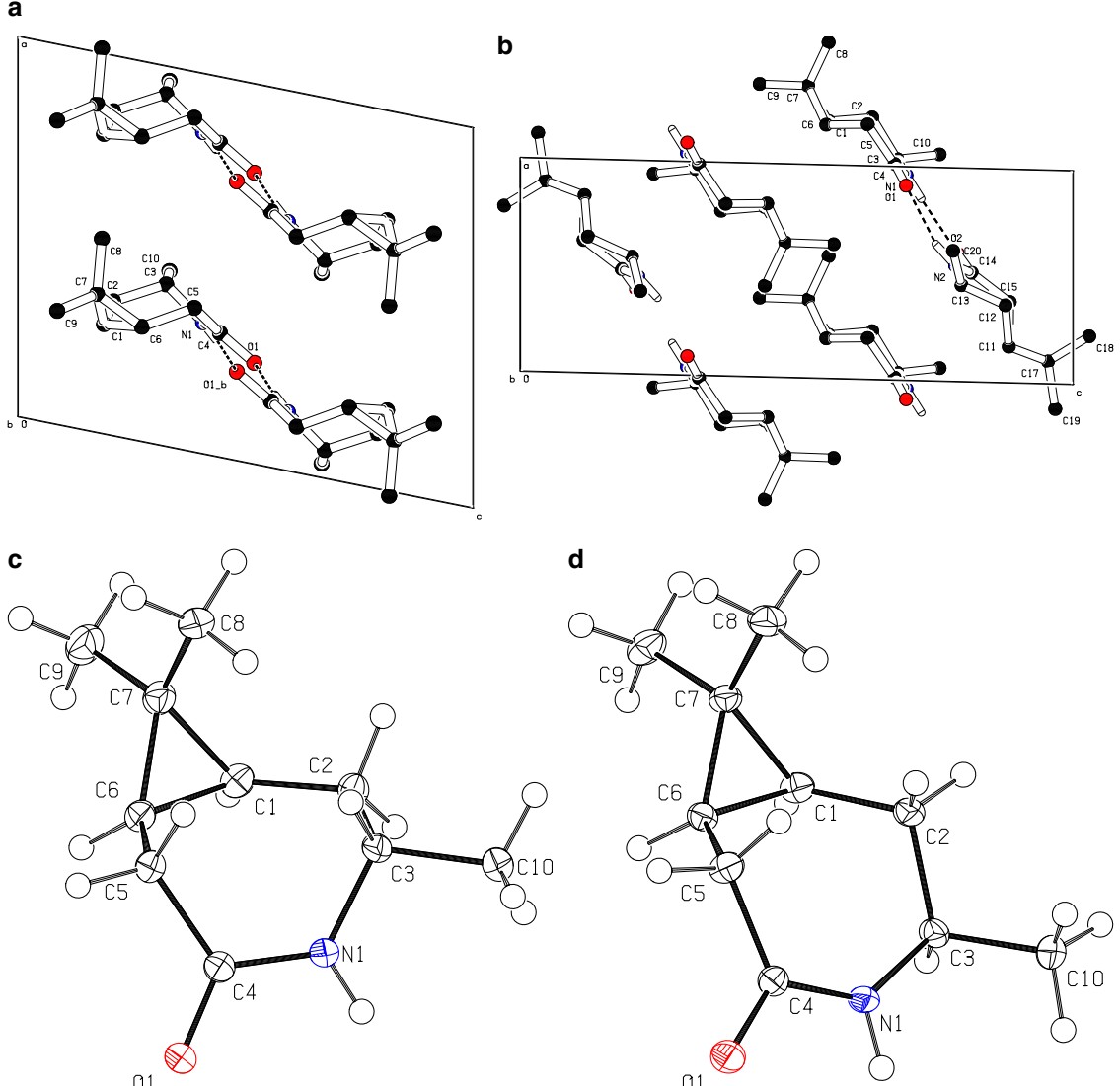

**Fig. 9 Crystal structures of the monomers.** Crystal structures and monomer molecule conformations of each **5-3R** (**a, c**) and **5-3S** (**b, d**) with bonding hydrogens only viewed in unit cell direction *b*.

104.49° is within a realistic margin to the ideal tetrahedron angle of 109.28°. The view of the expanded repeating unit demonstrates that the chain conformation is strongly influenced by the three-membered ring C1–C6–C7–C1 (Fig. 10). The crystal structure view in the same figure shows that the "bend" shape of the C2–C3–N1–C4O1–C5 segment results from the hydrogen bonds across chains, in conjunction with the angle imposed by the three-membered ring.

## Discussion

To summarize, a stereoselective reaction sequence for two stereoisomeric lactams—starting from the chiral biomolecule (+)-3-carene—was developed. The selective Meinwald rearrangement enables the deliberate synthesis of **5-3S** or **5-3R**—new bio-based building blocks for either amorphous (transparent) or semi-crystalline polyamides. **5-3S** can be produced in an at least partly sustainable one-vessel process without purification of any intermediates. The final product was effortlessly purified by crystallization. The initial oxidation step could be achieved by in situ generated peracetic acid, formed by a lipase, $H_2O_2$, and acetic acid, a considerably environmentally benign and safe method for epoxidation.

The results of our small-scale basic polymerization experiments show that **5-3S** and **5-3R** polymerize comparably facile and that the molecular weight can be adjusted by variation of the activator amount. Additionally, the general potential to form co-polyamides with CL and LL allows the formation of new partially bio-based materials. The homopolymers and the co-polymers have very intriguing characteristics, regarding thermal properties, crystallinity, and transparency. These properties can be attributed to the three-membered ring and the methyl group at C3, which—according to NMR analysis—maintain their configuration throughout the polymerization. Especially the co-polymers are of high interest as the properties of these new polymers highly differ from PA6 and PA12, possibly giving way to new fields of application without the need of additional additives to introduce similar effects. Furthermore, all produced polyamides are, by chemical logic, chiral as the starting material (+)-3-carene consists of only one enantiomer and the three-membered ring is unable to isomerize. This might enable applications in chiral separation techniques.

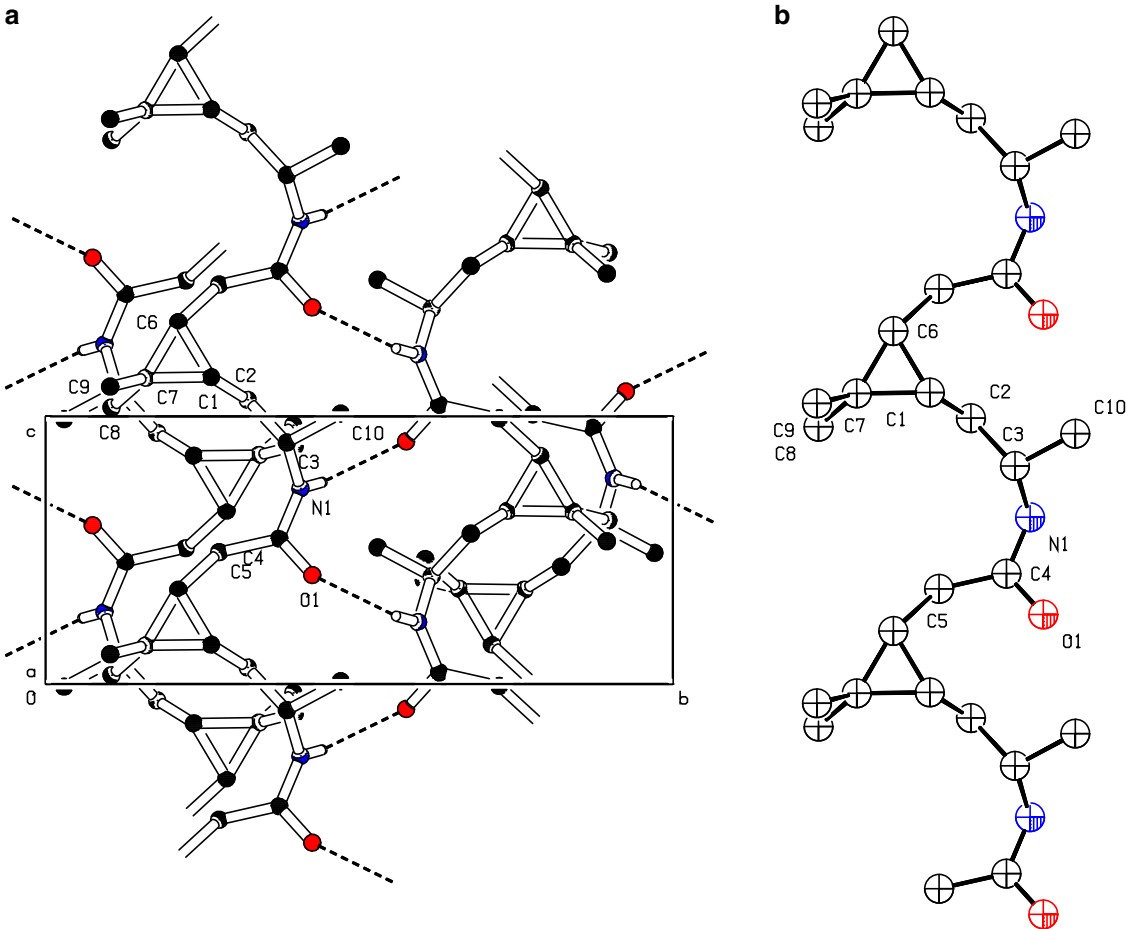

**Fig. 10 Crystal structure of poly5-3S.** Structure of **poly5-3S**, viewed in unit cell direction *a* (**a**) and the expanded repeating unit (**b**). In the unit cell view, only bonding hydrogens are shown.

In the future, we plan to scale-up the polymerization of the homo- and co-polymers, the investigation of the hydrolytic polymerization, compatibility tests with commercial poly-merization systems and additives, and polymer processing to produce standardized specimen to further demonstrate that high-performance bio-polyamides can not only keep up with their fossil-based counterparts but also that bio can be better.

## Methods
**Instrumental and characterization**. GCMS was performed using GC-2010 Plus (Shimadzu) with an auto-injector AOC-5000 (Jain, Combi PAL), a GC capillary column (BPX 5: 5% phenyl, 95% methyl polysilphenylene/siloxane; SGE), and MS-QO2010 Plus (Shimadzu) at 70 eV. NMR measurements were carried out on a JNM-ECA 400 MHz spectrometer from JEOL. Chemical shifts δ are indicated in parts per million with respect to residual solvent signals. SEC was performed using a SECcurity GPC system with an autosampler (1260 Infinity; Agilent Technologies) and a TCC6000 column oven (Polymer Standards Service, PSS). DSC was per-formed on a DSC 1 from Mettler Toledo with the software STARe V. 16.00. MALDI-TOF was conducted on a Bruker Ultra Flex TOF/TOF mass spectrometer. Single-crystal analysis was performed by single-crystal X-ray diffractometry (SC-XRD, D8 Venture, Bruker AXS, Madison, WI, USA) equipped with a 4-circle goniometer (Kappa geometry), a CMOS detector (Photon 100, Bruker AXS), a rotating anode (TXS, Bruker AXS) with MoKα radiation (λ = 0.71073 Å), and a multilayer mirror monochromator (HELIOS, Bruker AXS). Powder X-ray dif-fraction was carried out using Bragg-Brentano geometry (PXRD, Miniflex, Rigaku, Japan, with silicon strip detector D/teX Ultra) and copper Kα radiation. Detailed information about applied methods and sample preparation is described in the Supplementary Methods.

**Monomer synthesis**. All applied chemicals, including solvents, were purchased in industrial grade and used as received. All monomer synthesis reactions were car-ried out under air without inert atmosphere. For the enzymatic epoxidation,

Novozyme-435 (CALB lipase immobilized on acrylic resin) was applied. The reaction progress was monitored by GCMS or TLC. Purification of the products was achieved by distillation, crystallization, or column chromatography. Product characterization was realized by NMR and GCMS. The exact structure of **5-3S** and **5-3R** was determined by XRD. TLC was performed using aluminium plates coated with SiO$_2$ (Merck 60, F-254) and the spots were visualized with a KMnO$_4$ stain. Flash column was performed using SiO$_2$ (0.06–0.2 mm, 230–400 mesh ASTM) from Roth. NMR assignments of all intermediates are provided beneath the synthesis protocol, NMR data are displayed in Supplementary Figs. 37–46.

**One-vessel scale-up**. The reaction cascade was performed under application of the METTLER TOLEDO LaBMax Automatic Lab Reactor equipped with a 4.50 L reaction vessel with a bottom outlet, a stirrer (shaft stirrer blade), a condenser, a distillation bridge, and active water cooling (minimum temperature: 15 °C). A detailed step-by-step protocol is described in the Supplementary Methods.

**Polymerization method A**. 3S-caranalctam (**5-3S**), 3R-caranlactam (**5-3R**), **CL** or **LL** and the specific amount of activator (N-Bz-caranlactams **Bz5-3S** and **NBz5-3R**) and NaH (60% on paraffin) were put into a glass vial (10 mL) equipped with a magnetic stir bar. The vial was closed with a screw lid with a rubber septum and vacuum was applied via a syringe connected to a vacuum pump for 10 min before the vial was vortexed for 30 s. The vial was placed in an oil bath and stirred at a specific temperature. When the reaction time was over, the vial was removed from the oil bath and cooled down to room temperature without external cooling.

**Polymerization method B**. 3S-caranlactam (**5-3S**), 3R-caranlactam (**5-3R**), **CL** and **LL**, and the specific amount of activator (N-Bz-caranlactams **Bz5-3S** and **NBz5-3R**) and NaH (60% on paraffin) were put into a glass vial (10 mL), flushed with nitrogen and closed with a screw lid with a rubber septum and vortexed for 30 s. The vial was then placed into a heating block at different temperatures for the respective amount of time. The heating block was covered with aluminium foil to decrease the temperature gradient. After the reaction time was reached, the vial cooled down to room temperature without external cooling.

**Polymer work-up method A**. The glass vial was destroyed and glass residues sticking to the polymer were removed before it was broken down to small pieces under application of scissors, hammering and liquid nitrogen if necessary. The polymers from lactam **5-3S** were more brittle than the polymers from **5-3R**, whereas the poly-3R-caranlactam (**poly5-3R**) was at least partly soluble in EtOH. The polymer pieces were transferred to a mortar and grinded in the presence of a few millilitres of organic solvent (EtOAc for **poly5-3R**, EtOH for **poly5-3S**) until a fine powder was produced. In cases of non-homogeneous particles, the powder was refluxed in a mixture of EtOH and water (1:1) for at least 12 h, filtered off, and washed several times with water, acetone, and EtOAc and grinded again. Finally, the colourless or slightly yellow powders were dried under reduced pressure and analysed by IR, NMR, DSC, and GPC.

**Polymer work-up method B**. The polymers were dissolved in HFIP within the polymerization glass vial and a sample of the homogeneous solution was analysed by GPC. Another sample was dried under reduced pressure, re-dissolved in DCOOD and investigated by NMR. Specific NMR signals as well as the GPC elugrams were used for the determination of monomer conversion.

**Effect of temperature and activator concentration on M**. Lactam **53-S** was converted to **poly5-3S** at 180 and 220 °C at different concentrations of activator **Bz5-3S**. Monomer (300 mg, 1.8 mmol, 1.00 equiv.), NaH (60% on paraffin, 1.6–4.0 mg, 0.04–0.10 mmol, 0.02–0.05 equiv.), and activator **Bz5-3S** was polymerized for 1 h as described in polymerization method A. Polymer work-up A was used for further investigations. The results are displayed in Supplementary Figs. 8 and 9 and Supplementary Table 11.

**Impact of the activator concentration on the conversion**. Monomer (500 mg, 3.0 mmol, 1.00 equiv.), NaH (60% on paraffin, 3.5 mg, 0.09 mmol, 0.03 equiv.), and a varying amount of activator was polymerized at 190 °C for 1 h as described in polymerization method B. Polymer work-up B was used for further investigations. The results are displayed in Supplementary Figs. 11–16 and Supplementary Table 13.

**Impact of the reaction time on the conversion**. 3S-caranlactam (**5-3S**, 300 mg, 1.80 mmol, 1.00 equiv.), NaH (60% on paraffin, 6.0 mg, 0.15 mmol, 0.08 equiv.), and **Bz5-3S** (15.0 mg, 0.06 mmol, 0.03 equiv.) was polymerized by polymerization method B several times. Each polymerization experiment was terminated after a specific reaction time. The polymers were dissolved in a mixture of HFIP and EtOH, and a sample was dried under reduced pressure. The remaining residue was re-dissolved in DCOOD and analysed by NMR as displayed in Supplementary Fig. 20 and Supplementary Table 14.

**Copolymerizations of 3S-caranlactam (5-3S)**. Lactam **5-3S** was co-polymerized with **5-3R**, **LL**, and **CL** under various conditions as displayed in Supplementary Table 8. The ratio of lactam **5-3S** and the specific co-monomer (Built-in) of the polyamides copoly(**5-3S·5-3R**), copoly(**5-3S·LL**), and copoly(**5-3S·5-CL**) was investigated by NMR. The protons used for integration and comparison are highlighted in Supplementary Fig. 27; the results are displayed in Supplementary Figs. 23–25 and Supplementary Table 16. ¹H and ¹³C spectra showing the full ppm range are displayed for an example of each type of co-polyamide as Supplementary Figs. 47–49.

**Time-dependent integration of 5-3S in the CL/LL co-polymers**. The time-dependent integration of **5-3S**, **CL**, and **LL** in the growing co-polymer chain was investigated. 1:1 mixtures of the monomers were polymerized as follows: The monomers were melted in a round bottom flask equipped with a magnetic stirrer bar in a nitrogen atmosphere. NaH was added, followed by **Bz5-3S**. Samples from the melt were taken at different reaction times until the reaction mixture became solid. A final sample was taken from the solid after 60 min. The samples were analysed by NMR and the conversion and integration were determined (Supplementary Figs. 28, 29 and Supplementary Table 17).

**Film-cast experiments**. 0.5 g of PA6, PA12, **copoly(5-3S·CL)**, or **copoly(5-3S·LL)** were dissolved in HFIP (20 mL) for at least 12 h. The solution was filtered and transferred in a crystallizing dish (diameter: 11 or 5.5 cm) and left under a fume hood until a clear film was formed. The film was carefully separated from the dish and dried in an oven at 85 °C for 3 h to remove residual solvent. For **poly5-3R**, this method proved unsuitable as the dried polyamide film could not be separated from the glass. Therefore, a PTFE foil was used as an inlay. Films of **poly5-3R** could then be separated from the PTFE inlay after drying.

**Single-crystal production of the lactams**. Single crystals were obtained by crystallization at 4 °C within 3 days. The crystals were separated by filtration, washed with cold acetone, and dried under air atmosphere.

## Data availability
The authors declare that the data supporting the findings of this study are available within the article (and its Supplementary Information files). The X-ray crystallographic coordinates for structures reported in this study have been deposited at the Cambridge Crystallographic Data Centre (CCDC), under deposition numbers 1938732 (**5-3S**) and 1938733 (**5-3R**). These data can be obtained free of charge from The Cambridge Crystallographic Data Centre via www.ccdc.cam.ac.uk/data_request/cif. Crystallographic data of the polymers are available from Daniel Van Opdenbosch (daniel.van-opdenbosch@tum.de) at reasonable request.

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

## Acknowledgements

We would like to thank the Ministerium für Ernährung und Landwirtschaft (BMEL) and the Fachagentur Nachwachsende Rohstoffe e. V. (FNR) for generous funding (FKZ 22015916).

## Author contributions

P.N.S., M. Winnacker, H.S., and V.S. conceived the work. P.N.S., D.L.P., M.H., S.L., J.R., M. Woelbing, and C.F. performed the synthetic experiments and characterization. X-ray experiments, structure determinations, and related manuscript/supplementary sections were conducted by A.P. (monomers), D.V.O. (polymers), and C.Z. (polymers). P.N.S. wrote the manuscript and supplementary except X-ray sections by A.P. and D.V.O. C.Z. and V.S. revised the manuscript.

## Competing interests

The authors declare no competing interests.

**Additional information**

