## [Peer Review File · Nature Communications]

Reviewers' comments:

Reviewer #1 (Remarks to the Author):

This communication by Sieber and coworkers describes the synthesis of a new diastereomeric 3S-caranlactam and an optimized synthesis for the (+)-3-carene-based 3R-caranlactam. Interestingly, a four-step one-vessel monomer synthesis with an overall yield of 25 %, applying chemo-enzymatic catalysis for the initial oxidation step, was developed. These monomers were then polymerized to poly-3R-caranamide and poly-3S-caranamide, and the copolymerization of 3S-caranlactam with caprolactam and lauro lactam was also investigated.

The results obtained show that the molecular weight can be adjusted by variation of the activator amount. The authors also report that the polyamides and the copolyamides have unique high-performance properties, such as transparency, crystallinity, and thermal properties. For instance, the copolymers properties highly differ from PA6 and PA12, which open the way to new avenues. Based on the NMR spectroscopic analysis of macromolecular structures, the authors proposed that these properties can be attributed to the three-membered ring and the methyl group at C3, which maintain their configuration throughout the polymerization.

The ideas are logically presented and discussed. The writing is sufficiently clear to engage a broad, science-literate readership interested in polymer chemistry. Altogether, the scientific quality of this article is excellent since a lot of information is provided and it covers most of the recent literature devoted to this field. In summary, I recommend publication with few modifications.

1. In order to rationalize the polymerization mechanism, the authors should analyze some of their polymers by MALDI-TOF spectrometry.
2. As a minor correction, I would also suggest citing the following review article describing the main achievements in one-pot synthesis of (co)polymers: Chemical Society Reviews 2013, 42, 9392-9402

Finally, the topic of this article is timely and certainly relevant for the broad readership of Nature Communications. This is an excellent manuscript and I look forward to seeing it published in Nature Communications.

Reviewer #2 (Remarks to the Author):

A route to bioderived, terpene based polyamides is described. Renewable polyamides from such resources were described before, the novelty lies in the stereoselective synthesis and the thus resulting stereospecific polymers. However, chiral polyamides are also known and some of the authors published on the topic before (i.e. Macromol. Rapid Commun. 2016, 37, 851; doi: 10.1002/marc.201600056), thus limiting conceptual novelty.

A sustainable synthesis procedure is claimed, but neither 25% overall yield, nor the applied reaction conditions fulfil such criteria – for instance, acetonitrile is a very undesirable solvent in practically any solvent selection guide and NBS as well as the combination NaOH/TsCl result in very high amounts of waste. These are only the most obvious limitations in terms of sustainability. The authors should provide a sustainability metric and compare their synthesis to other routes, including all starting materials as well as solvents in their consideration. The E-Factor, for instance, will be extremely high, as also the required washing steps and solvent changes contribute to waste (a recycling is not demonstrated in the manuscript).

In summary, this manuscript does not show the novelty/urgency required for this journal and should be submitted to a journal in the field of polymer science.

Reviewer #3 (Remarks to the Author):

The authors report in this manuscript the homo- and co-polymerizations of two diastereomeric, 7-membered lactams, each derived from the terpene (+)-3-carene. This is a comprehensive study and the Supplementary Information is thorough and fully supportive of the conclusions reached in the manuscript. These studies build upon earlier work from the same laboratory in which one of the same isomeric monomers was synthesized and polymerized to its homopolymer (ref. 41 in 2019; a parallel study of a lactam made from pinene was reported there as well). The researchers now report: i) synthesis of both diastereomers (epimeric at the methyl-bearing stereocenter) by a route different and more efficient from that reported earlier, ii) an impressive telescopic and one-pot synthesis of one of the two isomers that is demonstrably amenable to large scale, iii) detailed studies of the synthesis and (complementary) properties of the stereoisomeric homopolymers derived from each of the monomers, and iv) synthesis of various copolymers between one of the carene-derived lactams and caprolactam (CL) or laurolactam (LL), through which the thermal and morphological properties of homo-PCL and homo-PLL can be modified in controlled fashion. Although the ref. 41 report compromises somewhat the impact of the work reported here, on balance I still find the advances now revealed to be sufficiently novel, important, and practically valuable to be appropriate for publication in Nature Comm. I have made comments below that the authors should consider in preparing an improved version of the manuscript.

To my taste there was, scattered throughout the text, too much speculation or forward-looking plans of what might be additionally true or what is being or should be studied next. While these were all valid points, I didn't find they added much to the messaging of the results in hand. The most extreme example of this appears in the second paragraph of the conclusions. By the time I finished reading there, I had to remind myself that I was reading a manuscript for publication and not a grant proposal application.

The efforts to optimize the chemoenzymatic step using deep eutectic solvents as replacements for ionic liquids, while fundamentally interesting, seem misguided from a preparative point of view. At the end of the day, this simply represents an alternative synthesis of peracetic acid (the outcomes using CALB perhydrolysis vs. MeCO_3H itself are, within error, identical—p 28 of the supporting material). The ultimate stoichiometric oxidant, H_2O_2 , is identical for the two routes, the use of the pricey enzyme is unlikely to ever be economically viable to scale-up, and the synthesis of peracetic acid from H_2O_2 and AcOH (vinegar) or acetic anhydride has very little green-downside. This is a different way of saying that I find the statement that this method is "a considerably environmentally benign and save method for epoxidation" (conclusion) is not particularly convincing nor warranted.

The copolymerizations with CL and LL each involve two monomers that likely undergo anionic ring-opening at considerable different rates. As such, the steepness of the gradient of the tapered structures of the resulting copolymers may well be quite steep. Analysis of aliquots of the reactions at intermediate degrees of polymerization for remaining monomer concentration would give a direct measurement of relative rates of monomer consumption.

The naming of diastereomeric compounds as #-3S and #-3R is curious. The methyl-bearing stereocenter in epoxides 2 is at C3, but in the ketones and oximes 3 and 4 it is at C4 and in lactams 5 at C5, so it is a bit odd that the last three are still labeled as #-3R or #-3S. That numbering is not correct and is only unambiguous in the knowledge of the origin of the synthesis and precursor in which that stereocenter is first introduced.

The following minor points were encountered as I read through the document.

"cyclic intramolecular amide which was first polymerized" to "cyclic amide that was first

polymerized"

"The latter three are the main components of turpentine oil ..." Please provide the ca. amount or range of carene typically found in turpentine.

"PA11 and PA12 and the" to "PA11 and PA12, and the"

correct "180°C"

correct "hexafluoro-2-propanol"

"the defined stereochemistry leads ... to chiral polyamides." to "the defined relative (and absolute) configuration gives rise ... to diastereomerically (and enantiomerically) pure polyamides."

"We figured" to "We presume"

"but for other" to "but has been reported for other"

"rigid three-membered ring in the polymer backbone and the resulting high conformational barriers." to "rigid three-membered ring in the polymer backbone and the resulting reduction in one degree of rotational freedom in the polymer backbone." It is not possible to rotate around the cyclopropane bond, so conformational issues are not in play. [BTW, substituents along an acyclic backbone have a "rigidifying" effect not because they slow down (much at all) the rate of conformational interconversion but because they perturb the equilibrium distribution/population of the ensemble of conformations that are accessed by the chains.]

Clarify "Copolymerization of 5-3S with 5-3R, CL and LL." to make it clear that these were all copolymerization between just two monomers, not all four at once.

"we figured that the random inclusion" to "we hypothesized that the random inclusion"

Clarify what is meant by (the non-conventional) "the measured built-in never reached 100% of the theoretical maximum built-in." (and one additional use of "built-in")

"monomers, and - due to" to "monomers and - due to"

Clarify what is meant by "strong sublimation"

"underpinning the partial double bond" to "reflecting the partial double bond"

"In case of the structure of 5-3R," to "In the case of 5-3R,"

"except from residual monomers" to "except those from residual monomers"

"save method for epoxidation." to "safe method for epoxidation."

Rebuttal to the Reviewers' comments to NCOMMS-19-26889-T

We would like to thank all three reviewers for their valuable input.

Reviewer #1 (Remarks to the Author):

This communication by Sieber and coworkers describes the synthesis of a new diastereomeric 3S-caranlactam and an optimized synthesis for the (+)-3-carene-based 3R-caranlactam. Interestingly, a four-step one-vessel monomer synthesis with an overall yield of 25 %, applying chemo-enzymatic catalysis for the initial oxidation step, was developed. These monomers were then polymerized to poly-3R-caranamide and poly-3S-caranamide, and the copolymerization of 3S-caranlactam with caprolactam and lauro lactam was also investigated.

The results obtained show that the molecular weight can be adjusted by variation of the activator amount. The authors also report that the polyamides and the copolyamides have unique high-performance properties, such as transparency, crystallinity, and thermal properties. For instance, the copolymers properties highly differ from PA6 and PA12, which open the way to new avenues. Based on the NMR spectroscopic analysis of macromolecular structures, the authors proposed that these properties can be attributed to the three-membered ring and the methyl group at C3, which maintain their configuration throughout the polymerization.

The ideas are logically presented and discussed. The writing is sufficiently clear to engage a broad, science-literate readership interested in polymer chemistry. Altogether, the scientific quality of this article is excellent since a lot of information is provided and it covers most of the recent literature devoted to this field. In summary, I recommend publication with few modifications.

1. In order to rationalize the polymerization mechanism, the authors should analyze some of their polymers by MALDI-TOF spectrometry.

Poly5-3S and poly5-3R were investigated by MALDI-TOF. The characteristic mass peaks were found, but only oligomers were detected (about 6.0 kDa). This is a known phenomenon for polyamides (see references in the manuscript).

The following paragraph was added:

GPC was chosen for determination of the molecular weight as measurements of poly5-3R and poly5-3S with MALDI-TOF did show the characteristic peak distance of 167 m/z, but only oligomers up to 6.5 kDa could be detected (Supplementary Section A9). This is a known phenomenon for polyamides.^{38,65,66}

2. As a minor correction, I would also suggest citing the following review article describing the main achievements in one-pot synthesis of (co)polymers: Chemical Society Reviews 2013, 42, 9392-9402

The review describes tandem catalysis as a method for (new) polymers. While we think that this interesting topic does not completely fit the content of the manuscript, reference 34 of the review deals with the synthesis of new polyesters, utilizing monoterpene epoxides. This perfectly matches the scope of our manuscript, and the reference was included.

Robert, C., Montigny, F. de & Thomas, C. M. Tandem synthesis of alternating polyesters from renewable resources. Nat. Commun. 2, 586; 10.1038/ncomms1596 (2011).

Finally, the topic of this article is timely and certainly relevant for the broad readership of Nature Communications. This is an excellent manuscript and I look forward to seeing it published in Nature Communications.

Reviewer #2 (Remarks to the Author):

A route to bioderived, terpene based polyamides is described. Renewable polyamides from such resources were described before; the novelty lies in the stereoselective synthesis and the thus resulting stereospecific polymers.

Apart from the stereoselective synthesis (which is realized by a Meinwald-rearrangement catalysis system that has not been reported before), novelties compared to other works dealing with the synthesis of polyamides from terpenes are as follows:

- Semi-crystalline or amorphous polyamides starting from the same feedstock which leads to
- Transparent or opaque materials
- Controlled polymerization leading to sufficiently high molecular weights and increased conversion and decreased reaction times
- Detailed investigation of the co-polymerization not only with another PA 6-type monomer (caprolactam) but also with a PA12-type monomer (lauro lactam), drastically changing the properties of regular PA6 or PA12
- Crystal structure of a terpene-based polyamide
- Scale-up of the monomer synthesis
- Facile product isolation (no stereo- or regio-isomers) without energy consuming methods (distillation) or material-intensive methods (chromatography) at any stage of the reaction sequence

However, chiral polyamides are also known and some of the authors published on the topic before (i.e. *Macromol. Rapid Commun.* 2016, 37, 851; doi: 10.1002/marc.201600056), thus limiting conceptual novelty.

Regarding the stereochemistry, the surprising highlight is not the chirality of the polyamides (although this offers possibilities in chiral technique applications for the homo- and the co-polyamides) but the fact that the epimerization of a single side-chain methyl group (in a repeating unit containing 10 carbon atoms) leads to a complete loss of long-range order. Poly-3R-caranamide and Poly-3S-caranamide are diastereomers and not enantiomers. This offers semi-crystalline and amorphous applications starting from the same raw material. This is not due to tacticity effects, as Poly-3S-caranamide and Poly-3R-caranamide are each isotactic. This also means that (+)-3-carene can be converted into amorphous or semi-crystalline polyamides without any additives or special processing. A similar concept has not been reported before for any polymer, so we strongly disagree with the reviewer's evaluation of the novelty of the presented results. The only similarity is the use of terpenes as raw material for lactams and the polymerization system.

A sustainable synthesis procedure is claimed, but neither 25% overall yield, nor the applied reaction conditions fulfil such criteria – for instance, acetonitrile is a very undesirable solvent in practically any solvent selection guide and NBS as well as the combination NaOH/TsCl result in very high amounts of waste.

We agree with the reviewer that the sustainability of our process still has potential for improvement in some aspects (especially the Beckmann rearrangement), but we also think that the sustainability of the process has to be considered in the context of the already existing synthesis of 5-3R by S. Lochynski (reference 43) and, for example, the synthesis of the bio-based polyamide 11 (40-50% polymer yield starting from castor oil, 67% monomer yield after decades of development, temperatures from -10 °C up to 550 °, application of HBr, requirement of very complex reactors, and so on).

The sustainable aspects of the presented synthesis are moderate temperatures, the product isolation without energy intensive/material intensive techniques, the metal-free synthesis with the exception of very little amounts of iron and the 100% green carbon of the polyamides.

It is true that NBS is not sustainable, but we only used it in pathway B to show that 2-3R can be converted to 3-3R by our catalytic system with high diastereomeric excess. 5-3R is also accessible from 3-3S and subsequent acidic isomerization as displayed in Figure 1. NBS can be avoided completely. NBS was also not part of the scaled reaction.

Acetonitrile (MeCN) is a non-nucleophilic, dipolar aprotic solvent, which is required for this type of reaction. According to various solvent selection guides, MeCN is the most sustainable solvent in this class (/Solvent guide = evaluation result /: AstraZeneca = undesirable, ACS Green Chemistry Institute Pharmaceutical Roundtable = usable, GSK = usable, Pfizer = usable, Sanofi = recommended). MeCN is not a perfectly green solvent, but the statement *“acetonitrile is a very undesirable solvent in practically any solvent selection guide”* appears to us as an exaggeration. In addition, MeCN is a byproduct of polyacrylonitrile and utilization of byproducts has also some merit in sustainability.

These are only the most obvious limitations in terms of sustainability. The authors should provide a sustainability metric and compare their synthesis to other routes, including all starting materials as well as solvents in their consideration. The E-Factor, for instance, will be extremely high, as also the required washing steps and solvent changes contribute to waste (a recycling is not demonstrated in the manuscript).

There are no *“other routes”* reported yet, except for the toxic, expensive, and non-scalable synthesis of 5-3R by S. Lochynski which is certainly not sustainable. 5-3S has not been prepared before.

As described in the supplementary information, all applied solvents are technical grade and the synthesis does not require highly purified solvents. Therefore, the cyclohexane that is separated by (azeotropic) distillation can be re-used. The same is true for the ethyl acetate used for the product crystallization and even acetonitrile, which is also present in the mother liquid, could be isolated by distillation and used again.

To address the issue of the synthesis being just more sustainable compared to previously described ones but not fully sustainable we decreased the focus on the sustainability of the synthesis process within the manuscript and added following paragraph:

“Although the synthesis is not fully optimized, several sustainable aspects are worth mentioning. The presented process requires only moderate reaction conditions and no elaborated reaction equipment.

Only little amounts of metal are used during the process. In addition – as no low-boiling, interfering side products are formed – cyclohexane is retrieved during the process and can be reused; EtOAc and MeCN can also be isolated from the mother liquor by distillation after product crystallization. Finally, the product is purified by crystallization in polymerization-grade, requiring neither material- or energy-consuming methods. However, for a fully sustainable synthesis the amount of washing solutions – which were used in great excess so far – must be reduced, a catalytic method for the Beckmann rearrangement needs to be implemented, and an increase of the overall yield is required.”

In summary, this manuscript does not show the novelty/urgency required for this journal and should be submitted to a journal in the field of polymer science.

Reviewer #3 (Remarks to the Author):

The authors report in this manuscript the homo- and co-polymerizations of two diastereomeric, 7-membered lactams, each derived from the terpene (+)-3-carene. This is a comprehensive study and the Supplementary Information is thorough and fully supportive of the conclusions reached in the manuscript.

These studies build upon earlier work from the same laboratory in which one of the same isomeric monomers was synthesized and polymerized to its homopolymer (ref. 41 in 2019; a parallel study of a lactam made from pinene was reported there as well).

The researchers now report: i) synthesis of both diastereomers (epimeric at the methyl-bearing stereocenter) by a route different and more efficient from that reported earlier, ii) an impressive telescopic and one-pot synthesis of one of the two isomers that is demonstrably amenable to large scale, iii) detailed studies of the synthesis and (complementary) properties of the stereoisomeric homopolymers derived from each of the monomers, and iv) synthesis of various copolymers between one of the carene-derived lactams and caprolactam (CL) or laurolactam (LL), through which the thermal and morphological properties of homo-PCL and homo-PLL can be modified in controlled fashion. Although the ref. 41 report compromises somewhat the impact of the work reported here, on balance I still find the advances now revealed to be sufficiently novel, important, and practically valuable to be appropriate for publication in Nature Comm. I have made comments below that the authors should consider in preparing an improved version of the manuscript.

To my taste there was, scattered throughout the text, too much speculation or forward-looking plans of what might be additionally true or what is being or should be studied next. While these were all valid points, I didn't find they added much to the messaging of the results in hand. The most extreme example of this appears in the second paragraph of the conclusions. By the time I finished reading there, I had to remind myself that I was reading a manuscript for publication and not a grant proposal application.

We shortened the description of some of the future plans in the conclusion, but as this is a communication we think that the presentation of the work that will be done soon (or is already in progress) should also be described.

The efforts to optimize the chemoenzymatic step using deep eutectic solvents as replacements for ionic liquids, while fundamentally interesting, seem misguided from a preparative point of view. At the end of the day, this simply represents an alternative synthesis of peracetic acid (the outcomes using CALB perhydrolysis vs. MeCO₃H itself are, within error, identical—p 28 of the supporting material). The ultimate stoichiometric oxidant, H₂O₂, is identical for the two routes, the use of the pricey enzyme is unlikely to ever be economically viable to scale-up, and the synthesis of peracetic acid from H₂O₂ and AcOH (vinegar) or acetic anhydride has very little green-downside. This is a different way of saying that I find the statement that this method is “a considerably environmentally benign and save method for epoxidation” (conclusion) is not particularly convincing nor warranted.

This could be a misunderstanding, as we do not use deep-eutectic solvents in this study (but quote another study from our group which deals with this topic). The application of CALB and H₂O₂ can increase the safety, as only little amounts of MeCO₃H are present at any point of the reaction as it is formed *in-situ* and reacts very fast with the double bond; a potentially dangerous accumulation of peracetic acid is highly unlikely. The text was edited accordingly.

“The metal free enzymatic method generates epoxides in high yields under mild conditions, can be conducted in green solvents such as ethyl acetate, and prevents the potentially dangerous aggregation of peracetic acid.”

The copolymerizations with CL and LL each involve two monomers that likely undergo anionic ring-opening at considerable different rates. As such, the steepness of the gradient of the tapered structures of the resulting copolymers may well be quite steep. Analysis of aliquots of the reactions at intermediate degrees of polymerization for remaining monomer concentration would give a direct measurement of relative rates of monomer consumption.

The incorporation of caprolactam, laulolactam and 5-3S at intermediate degrees of polymerization (samples during the polymerization progress) was investigated by NMR. We chose NMR analysis of the growing polymer chains for this experiment as the very small changes of monomer concentration at the initial stages of polymerization (short reaction times) might lead to considerable errors. The results are described in the manuscript and the details are included in the Supplementary Information.

Following paragraph was added:

Investigation of the co-polymerization of 1:1 mixtures of 5-3S and CL/LL revealed that the monomers were consumed at different rates. The co-polymerization of 3-5S and CL almost exclusively starts with the conversion of CL before 3-5S is also incorporated (Supplementary section 9). Some examples for intermediate poly(5-3S-CL) were: poly(5-3S_{18%}-CL_{72%}) at 60 s (6.7% total conversion), poly(5-3S_{37%}-CL_{63%}) at 240 s (41% total conversion) and poly(5-3S_{49%}-CL_{51%}) at 60 min (82% total conversion). The co-polymerization with LL proceeded vice versa, and the co-polymer composition was poly(5-3S_{66%}-LL_{34%}) at 60 s (7.9% total conversion), poly(5-3S_{58%}-LL_{42%}) at 240 s (34.5% total conversion) and poly(5-3S_{52%}-LL_{48%}) at 60 min (64.4% total conversion). From this, it can be concluded that the relative rates of consumption are CL > 5-3S > LL.

The naming of diastereomeric compounds as #-3S and #-3R is curious. The methyl-bearing stereocenter in epoxides 2 is at C3, but in the ketones and oximes 3 and 4 it is at C4 and in lactams 5 at C5, so it is a bit odd that the last three are still labeled as #-3R or #-3S. That numbering is not correct and is only unambiguous in the knowledge of the origin of the synthesis and precursor in which that stereocenter is first introduced.

While this is true if IUPAC nomenclature is used, we named the compounds based on the suggestions of Mildred W Grafflin in *System of Nomenclature For Terpene Hydrocarbons Acyclics, Monocyclics, Bicyclics* (Advances in Chemistry 1955, Vol 14, Chapter 1, pp. 1-11, DOI: 10.1021/ba-1955-0014.ch001), which are often used in carbon labelling of bicyclic terpenes and their derivatives. We actually think that using this

nomenclature, the naming of the compounds and the stereocenter is less confusing. An explanation and the reference were added in the description of figure 1.

"The labelling of the stereo-centre C3 at all intermediates follows the recommendation for terpene carbon skeleton numbering of M. W. Grafflin, which suggest that the initial carbon labels of (+)-3-carene are fixed also in case of functionalization."⁴³ "

The following minor points were encountered as I read through the document.

"cyclic intramolecular amide which was first polymerized" to "cyclic amide that was first polymerized"

corrected

"The latter three are the main components of turpentine oil ..." Please provide the ca. amount or range of carene typically found in turpentine.

Paragraph ... "primarily isolated from the kraft pulping process (sulfate turpentine, 200 kt) or by distillation of resins extracted from conifers (gum turpentine, 100 kt). The composition highly depends on the species and origin of the utilized conifers. In Southeast USA, α -pinene (60-75%) and β -pinene (20-25%) are more common, whereas turpentine from Scandinavia and Russia contains considerable amounts of (+)-3-carene (40%)."²¹ " was added.

"PA11 and PA12 and the" to "PA11 and PA12, and the"

corrected

correct "180°C"

corrected

correct "hexafluoro-2-propanol"

corrected

"the defined stereochemistry leads ... to chiral polyamides." to "the defined relative (and absolute) configuration gives rise ... to diastereomerically (and enantiomerically) pure polyamides."

corrected

"We Figured" to "We presume"

corrected

"but for other" to "but has been reported for other"

corrected

"rigid three-membered ring in the polymer backbone and the resulting high conformational barriers." to "rigid three-membered ring in the polymer backbone and the resulting reduction in one degree of

rotational freedom in the polymer backbone.” It is not possible to rotate around the cyclopropane bond, so conformational issues are not in play.

corrected

[BTW, substituents along an acyclic backbone have a “rigidifying” effect not because they slow down (much at all) the rate of conformational interconversion but because they perturb the equilibrium distribution/population of the ensemble of conformations that are accessed by the chains.]

We thank the reviewer for this valuable correction.

Clarify “Copolymerization of 5-3S with 5-3R, CL and LL.” to make it clear that these were all copolymerization between just two monomers, not all four at once.

Description of Figure 8 was clarified.

“we figured that the random inclusion” to “we hypothesized that the random inclusion”

corrected

Clarify what is meant by (the non-conventional) “the measured built-in never reached 100% of the theoretical maximum built-in.” (and one additional use of “built-in”)

changed to “a complete consumption of 5-3S was not achieved”

“monomers, and – due to” to “monomers and – due to”

corrected

Clarify what is meant by “strong sublimation”

changed to “increased monomer sublimation was detected during the heating process”

“underpinning the partial double bond” to “reflecting the partial double bond”

corrected

“In case of the structure of 5-3R,” to “In the case of 5-3R,”

corrected

“except from residual monomers” to “except those from residual monomers”

corrected

“save method for epoxidation.” to “safe method for epoxidation.”

corrected

REVIEWERS' COMMENTS:

Reviewer #1 (Remarks to the Author):

As mentioned in my previous review, the authors present an interesting, original and potentially useful reaction. Data presentation, referencing and methodology is fine. This is solid piece of work and should be published. The authors have obviously taken criticisms and suggestions very seriously and accordingly have made significant revisions to the manuscript. I am satisfied that they have addressed some of my earlier comments on the preliminary nature of the mechanistic studies. I also believe that these corrections shall satisfy the concern expressed by the other reviewers.

I am convinced that Nature Communications is the appropriate forum for this work and I think that the manuscript proposed by the authors is sufficient for justifying publication in Nature Communications.

Reviewer #3 (Remarks to the Author):

In the response letter and revised manuscript by Sieber et al., all of the concerns and suggestions raised by the earlier three reviewers are addressed. The responses are balanced, objective, and persuasive; they have led to additions or changes that improve the manuscript and enhance its impact. One of the major concerns raised by the one dissenting Reviewer (#2) focused heavily on sustainability issues. While it is true that this work is likely not economically viable or ideally sustainable (incidentally, a criticism that can be applied to nearly all contributions to this field), there are, nonetheless, valuable conceptual features of the work that advance the field of polymer science more broadly. I remain supportive of publication.

The following are a few minor issues in the revised manuscript that should be easy to handle.

"elaborated reaction equipment." to "elaborate reaction equipment."

"can also be isolated from" to "can also be recovered from"

"crystallization in polymerization-grade," seems incomplete

I would still encourage that that term "rigid" (and the associated discussion) be changed in the manuscript text. This is a common misconception that novices and experts alike often bring to conformational analysis, and words like "rigid" or "frozen" perpetrate this misunderstanding. (Entirely coincidentally, I had a conversation about this very same issue with a polymer scientist a few days ago on a seminar visit.)

Reviewer #3 (Remarks to the Author):

In the response letter and revised manuscript by Sieber et al., all of the concerns and suggestions raised by the earlier three reviewers are addressed. The responses are balanced, objective, and persuasive; they have led to additions or changes that improve the manuscript and enhance its impact. One of the major concerns raised by the one dissenting Reviewer (#2) focused heavily on sustainability issues. While it is true that this work is likely not economically viable or ideally sustainable (incidentally, a criticism that can be applied to nearly all contributions to this field), there are, nonetheless, valuable conceptual features of the work that advance the field of polymer science more broadly. I remain supportive of publication.

The following are a few minor issues in the revised manuscript that should be easy to handle.

“elaborated reaction equipment.” to “elaborate reaction equipment.”

corrected

“can also be isolated from” to “can also be recovered from”

corrected

“crystallization in polymerization-grade,” seems incomplete

changed to “Finally, the product purity that is required for polymerization is reached by crystallization, avoiding material- or energy-consuming methods”

I would still encourage that that term “rigid” (and the associated discussion) be changed in the manuscript text. This is a common misconception that novices and experts alike often bring to conformational analysis, and words like “rigid” or “frozen” perpetrate this misunderstanding. (Entirely coincidentally, I had a conversation about this very same issue with a polymer scientist a few days ago on a seminar visit.)

“rigid” was deleted and some parts of the discussion were changed or deleted to prevent misunderstandings in conformational analysis. .